# Modelling response strategies for controlling *gonorrhoea* outbreaks in men who have sex with men in Australia

Qibin Duan[1,2], Chris Carmody[3,4], Basil Donovan[2,5], Rebecca J. Guy[2], Ben B. Hui[2], John M. Kaldor[2], Monica M. Lahra[6,7], Matthew G. Law[2], David A. Lewis[8,9,10], Michael Maley[11,12], Skye McGregor[2], Anna McNulty[5,13], Christine Selvey[14‡], David J. Templeton[2,15], David M. Whiley[16], David G. Regan[2‡], James G. Wood[13‡*]

1 School of Mathematical Sciences, Queensland University of Technology, Brisbane, Australia, 2 The Kirby Institute, UNSW Sydney, Sydney, Australia, 3 Liverpool Sexual Health Clinic, South Western Sydney Local Health District, Sydney, Australia, 4 Western Sydney University, Sydney, Australia, 5 Sydney Sexual Health Centre, South Eastern Sydney Local Health District, Sydney, Australia, 6 Microbiology Department, New South Wales Health Pathology, The Prince of Wales Hospital, Sydney, Australia, 7 School of Medical Sciences, UNSW Sydney, Sydney, Australia, 8 Western Sydney Sexual Health Centre, Western Sydney Local Health District, Sydney, Australia, 9 Westmead Clinical School, Faculty of Health and Medicine & Marie Bashir Institute for Infectious Diseases and Biosecurity, University of Sydney, Sydney, Australia, 10 Division of Medical Virology, Department of Pathology, Faculty of Health Sciences, University of Cape Town, Cape Town, South Africa, 11 Department of Microbiology and Infectious Diseases, Liverpool Hospital, Sydney, Australia, 12 South Western Clinical School, UNSW Sydney, Sydney, Australia, 13 School of Population Health, UNSW Sydney, Sydney, Australia, 14 Communicable Diseases Branch, Health Protection NSW, Sydney, Australia, 15 Department of Sexual Health Medicine, Sydney Local Health District and Discipline of Medicine, Central Clinical School, Faculty of Medicine and Health, The University of Sydney, Sydney, Australia, 16 Centre for Clinical Research, University of Queensland, Brisbane, Australia

‡ DGR and JGW are joint senior authors on this work. CS is Unavailable.
* james.wood@unsw.edu.au

**Data Availability Statement:** All relevant data are within the manuscript and its Supporting Information files.

## Abstract

The ability to treat gonorrhoea with current first-line drugs is threatened by the global spread of extensively drug resistant (XDR) *Neisseria gonorrhoeae* (NG) strains. In Australia, urban transmission is high among men who have sex with men (MSM) and importation of an XDR NG strain in this population could result in an epidemic that would be difficult and costly to control. An individual-based, anatomical site-specific mathematical model of NG transmission among Australian MSM was developed and used to evaluate the potential for elimination of an imported NG strain under a range of case-based and population-based test-and-treat strategies. When initiated upon detection of the imported strain, these strategies enhance the probability of elimination and reduce the outbreak size compared with current practice (current testing levels and no contact tracing). The most effective strategies combine testing targeted at regular and casual partners with increased rates of population testing. However, even with the most effective strategies, outbreaks can persist for up to 2 years post-detection. Our simulations suggest that local elimination of imported NG strains can be achieved with high probability using combined case-based and population-based test-and-treat strategies. These strategies may be an effective means of preserving current treatments in the event of wider XDR NG emergence.

**Funding:** DGR, JGW, BD, DMW and DAL received funding from the Australian National Health and Medical Research Council (NHMRC) under grant number APP1145642. JGW, DR, ML, MM, DL, BD, CC, DMW, CS, JK, SM, BH, QD received funding from a SPHERE III seed grant awarded in 2018. QD received salary support from APP1145642 and the III seed grant. The funders had no role in study design, data collection and analysis, decision to publish, or preparation of the manuscript.

**Competing interests:** The authors have declared that no competing interests exist. Author Christine Selvey was unable to confirm their authorship contributions. On their behalf, the corresponding author has reported their contributions to the best of their knowledge.

## Author summary

In most high-income settings, gonorrhoea is endemic among men who have sex with men (MSM). While gonorrhoea remains readily treatable with antibiotics, there are major concerns about the threat of antimicrobial resistance arising from recent reports of treatment failure with first-line therapy and limited remaining treatment options. Here we investigated the potential for test-and-treat response strategies to eliminate such strains before their prevalence reaches a level requiring a shift to new first line therapies. Rather than directly consider resistance, we explore the mitigating effect of various test-and-treat measures on outbreaks of a generic imported strain which remains treatable. This is done within the framework of a realistic mathematical model of gonorrhoea spread in an MSM community that captures cases, anatomical sites of infection and sexual contacts at an individual level, calibrated to relevant Australian epidemiological data. The results indicate that strategies such as partner testing and treatment in combination with elevated asymptomatic community testing are highly effective in mitigating outbreaks but can take up to 2 years to achieve elimination. As there are currently no clear alternative drugs of proven efficacy and safety to replace ceftriaxone in first-line therapy, these promising results suggest potential for use of these outbreak response strategies to preserve current treatment recommendations.

## Introduction

Gonorrhoea is a sexually transmissible infection (STI) caused by the bacterium *Neisseria gonorrhoeae* (NG). Antibiotics have, for many decades, provided effective treatment for gonorrhoea. However, resistance to several classes of antimicrobial agents has emerged, rendering many previously effective drugs ineffective [1]. Ceftriaxone is now recommended in most countries as the backbone of first line therapy for gonorrhoea. Since 2009, reports of NG strains exhibiting resistance to ceftriaxone have generated substantial concern in national and global public health agencies [2,3]. These reports were initially sporadic but by 2017 evidence emerged of sustained spread of ceftriaxone-resistant strains harbouring a novel resistance mechanism in the form of a *penA* type 60.001 allele [4]. Since then, a further extensively drug resistant (XDR) strain harbouring the *penA* 60.001 allele as well as high-level resistance to azithromycin has been reported in Australia, the UK and continental Europe [5–8]. With the possible exception of spectinomycin and gentamicin [9], there are no alternative drugs of proven safety and efficacy currently available for routine treatment of anogenital gonorrhoea. Novel antimicrobials are being trialled but are yet to be comprehensively assessed [10,11].

Without effective treatment and/or surveillance, there is the potential for rapid spread of resistance within populations as highlighted in South Australia in 2016, where azithromycin resistance rose from 5% to 30% of isolates within just 12 weeks, before dropping back to 12% in 2017 [12]. Rapid rises in ciprofloxacin resistance in New South Wales, Australia between 1991 and 1997 [13] and in South Africa in 2003 [14] have also been reported. These examples, combined with the recently reported cases of XDR NG, suggest that larger outbreaks of these or similar XDR strains are imminent, potentially arriving in Australia via repeated importation from the Asia-Pacific region as has been observed previously [13,15].

The prevalence of gonorrhoea in Australia is highest in remote Aboriginal and Torres Strait Islander populations and among men who have sex with men (MSM) in metropolitan centres. Although to date most reported cases of XDR NG have involved heterosexual contact

[5,7,8,16], we have chosen to focus on MSM due to the high incidence of gonorrhoea in this population, frequent sexual contact when travelling overseas, and evidence of the importance of oropharyngeal NG, which is more difficult to treat, in driving transmission in this population [17,18]. These factors together suggest that establishment of an XDR NG strain within an MSM population could spark a rapidly expanding gonorrhoea epidemic that would be difficult and costly to contain [19].

Without an effective gonococcal vaccine, developing appropriate public health responses to identify cases, reduce onward transmission, and maintain effective treatment for NG infections, are now key strategic priorities. However, beyond further changes to recommended antibiotic regimens, there is scant published evidence to inform the design of such responses. In this study we develop an anatomic site-specific individual-based model of gonorrhoea transmission in an urban Australian MSM population. Although similar in some aspects to other site-specific models of NG in MSM (e.g., two Australian studies [18,20] and two US studies [21,22]), this model has been specifically developed to examine outbreak control strategies for imported NG infection. Here, we evaluate the potential impact of community and individual-level test-and-treat strategies to control outbreaks of imported NG strains in the Australian MSM population.

## Results

### Model simulation and calibration

An individual-based model was developed that captures the dynamic formation and dissolution of sexual partnerships in a population of 10,000 MSM, sexual acts within partnerships, and the transmission of NG between 3 anatomical sites: urethra, oropharynx, anorectum. The natural history of gonorrhoea is captured in a Susceptible->Exposed->Infectious->Recovered->Susceptible (SEIRS) framework. Parameter values relating to gonorrhoea natural history have been derived, where possible, from published literature and are listed in Table 1. In each daily simulation cycle (illustrated in Fig 1), transmission events are tracked and the infectious status of all individuals updated. Events relating to natural progression of infection, testing, treatment of infection and entry/exit of individuals from the sexually active population are then processed before concluding each simulation cycle with partnership formation and dissolution.

The model was calibrated to estimated anatomical site-specific NG prevalence in a hypothetical community sample as reported in Zhang et al. [18]: oropharynx 8.6%; anorectum 8.3%; urethra 0.26% (Table 1). Comparison of 50 simulations from the calibrated model to prevalence targets is shown in Fig 2. Dynamic equilibrium prevalence is reached at approximately three years, with the site-specific prevalence curves then fluctuating around the target values. Figs A and B in S1 Text provide validation that the characteristics of the model-generated sexual contact network are consistent with data reported in the Gay Community Periodic Surveys (GCPS) Sydney 2018 [23] and the Health in Men (HIM) Study [24].

### Impact of outbreak response strategies

**Outbreak response strategies.**   The calibrated model is used to assess the effectiveness of intervention strategies in controlling strain importations. To simulate importation/emergence, we first select an infection site based on relative site-specific incidence rates reported in Callander et al. [25], yielding an oro-pharynx: urethra: rectum imported case ratio of 47:23:30. An imported gonococcal strain is then seeded in a randomly selected individual, already infected with the endemic strain, at the selected anatomical site. The endemic strain is removed immediately prior to strain importation, with simulations focusing on outbreaks related to the

**Table 1. Gonorrhoea infection parameters and prevalence targets for model calibration.**

| Parameter description | Value | Source |
|---|---|---|
| *Proportion of infections that are symptomatic by anatomical site* | | |
| Oropharyngeal infection | 0 | Oropharyngeal gonorrhoea is rarely associated with symptoms [44] |
| Urethral infection | 90% | [45,46] |
| Anorectal infection | 12% | [47] |
| *Average duration of infection stages (range)* | | |
| Oropharyngeal infection | 84 days (70,138) | Sampled from $\Gamma(201,0.4)$ within the specified range [48,49] |
| Urethral infection | 84 days (70, 140) | Sampled from $\Gamma(206,0.4)$ within the specified range [17,18] |
| Anorectal infection | 343 days (336,361) | Sampled from $\Gamma(3702,0.1)$ within the specified range [49] |
| Time from onset of anorectal/urethral symptoms to treatment | 3 days (1,7) | Sampled from $\Gamma(3,0.86)$ within the specified range [50] |
| Incubation | 4 days (2,10) | Sampled from $Exp(4)$ within the specified range [51] |
| Exposed | 3.6 days (1,9) | Sampled from $U(1,$length of incubation period$)$ within the specified range |
| Immunity | 3.5 days (1,7) | Assumption based on [52,53] |
| *Prevalence targets for model calibration (95% CI)* | | |
| Oropharynx | 8.6% (7.7–9.5) | Based on [18] and comparable with [54] |
| Anorectum | 8.3% (7.4–9.1) | |
| Urethra | 0.26% (0.04–0.35) | Based on [18] |

imported strain. We assume the imported strain is detectable and treatable and is not resistant to current treatment or alternatives. The intention, however, is to assess the effectiveness of test-and-treat strategies should a resistant strain emerge, assuming that last resort treatments such as carbapenems will be available for effective (potentially hospital-based) treatment.

We consider two levels of STI testing coverage (Table F in S1 Text): 1) current testing (CT) reflects testing coverage as reported in GCPS Sydney 2018 [23] and The Australian Collaboration for Coordinated Enhanced Sentinel Surveillance of Blood Borne Viruses and Sexually Transmitted Infections (ACCESS) [26]; and 2) recommended testing (RT) based on the 2016 Australian STI Management Guidelines [27] according to which all MSM should test for STIs at least once per year and those with 20+ sexual partners per year should test every three months. As a sensitivity analysis, we also provide results in the S1 Text for STI testing rates based on the 2019 Sexually Transmissible Infections in Gay Men Action Group (STIGMA) guidelines 2019 [28], which recommend 3-monthly testing for most sexually active MSM not in monogamous relationships. We assume testing occurs at all anatomical sites simultaneously with 100% sensitivity, 100% treatment efficacy in individuals who test positive for gonorrhoea infection, and clearance of viable gonococci from all anatomical sites within 1 day. Results for an alternative scenario assuming 95% test sensitivity, 95% treatment efficacy, and a 7-day clearance delay for asymptomatic infection are presented in the S1 Text.

Further, we examine the effect of combining different levels of sexual contact-based testing and treatment in with current/recommended screening. The Australian Contact Tracing Guidelines [29] recommend that recent (last 2 months) sexual contacts of index patients with gonorrhoea should be offered testing and treatment to minimise reinfection and onward transmission. We consider provision of testing/treatment to 80% or 100% of current regular

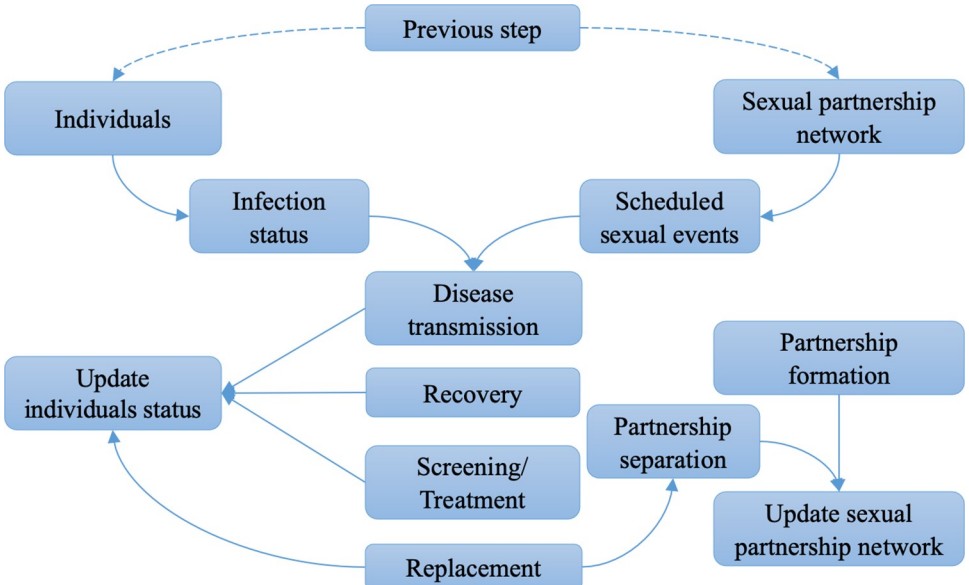

**Fig 1. Schematic illustration of sequence of events that occur in a single daily simulation cycle of the individual-based model.** In each cycle, the status of each individual and the sexual partnership network are carried over from the previous simulation step. Sexual events are scheduled daily for each partnership, and if disease transmission occurs, infection status is changed for the relevant individual(s). Events relating to natural progression of infection ("Disease transmission" and "Recovery"), testing, treatment of infection ("Screening/Treatment") and entry/exit of individuals ("Replacement") from the sexually active population are then processed before concluding each simulation cycle with partnership formation and dissolution for the next simulation cycle.

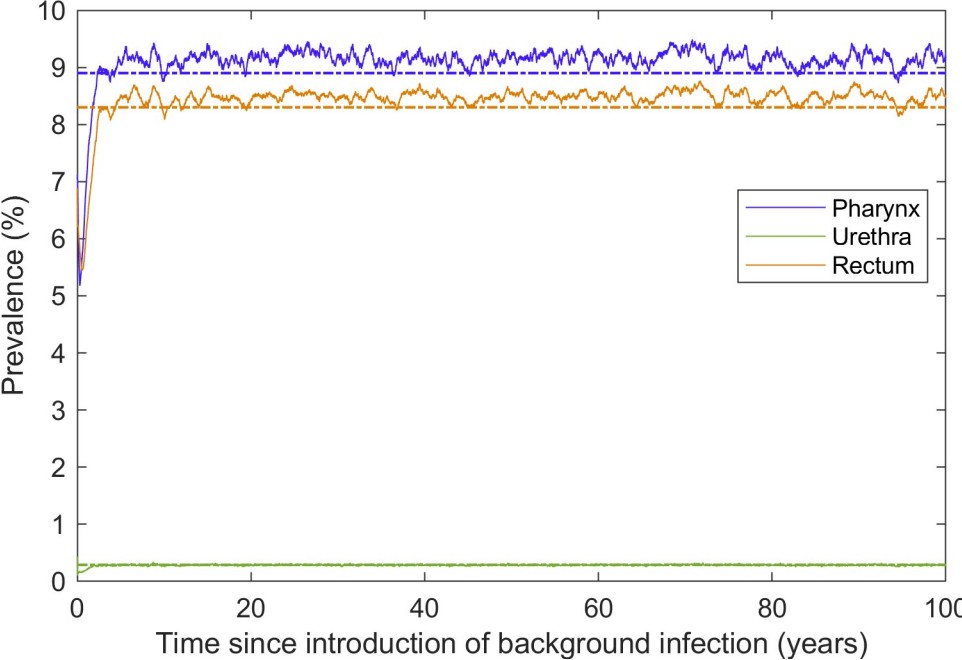

**Fig 2. Average daily model-generated site-specific prevalence over 50 simulations (solid lines) and calibration targets (dashed lines).** Each simulation was run for 100 years using the per-act transmission probabilities obtained through the calibration process and daily site-specific prevalence was averaged over the 50 simulations.

**Table 2. Outbreak response strategies and notation used.**

| Strategy | Current testing level (CT) | Recommended testing level (RT) |
|---|---|---|
| No contact tracing | CT | RT |
| Testing and treating 80% regular partners (PTT$_{R80}$) | CT +PTT$_{R80}$ | RT + PTT$_{R80}$ |
| Testing and treating 100% regular partners (PTT$_R$) | CT +PTT$_R$ | RT + PTT$_R$ |
| Testing and treating regular + 20% casual partners (PTT$_{RC20}$) | CT + PTT$_{RC20}$ | RT + PTT$_{RC20}$ |
| Testing and treating regular + 30% casual partners (PTT$_{RC30}$) | CT + PTT$_{RC30}$ | RT + PTT$_{RC30}$ |
| Testing and treating regular + 40% casual partners (PTT$_{RC40}$) | CT + PTT$_{RC40}$ | RT + PTT$_{RC40}$ |
| Testing and treating regular + 50% casual partners (PTT$_{RC50}$) | CT + PTT$_{RC50}$ | RT + PTT$_{RC50}$ |

partners (PTT$_{R80}$ and PTT$_R$) and four additional strategies, whereby 20%, 30%, 40% or 50% of casual partners in the last 2 months are tested/treated (PTT$_{RC20}$, PTT$_{RC30}$, etc.) as summarised in Table 2.

**Outbreak containment.** The model-predicted impact of outbreak response strategies on the ability of the imported strain to persist in the population are summarised in Table 3. Under the current reported level of testing (CT), the imported strain becomes extinct in 66% of simulations at 6 months post-importation, with a further 26% identified but persisting and 8% of outbreaks yet to be detected. By 2 years post-importation, 16% of simulated outbreaks persist, with mean population prevalence of 2.3% (IQR [0.5%-3.6%]) at any anatomical site. Almost 90% of this subset persist at 5 years post-importation, reflecting endemic establishment (mean prevalence 14.7% (IQR [14.3%-16.7%])). Prevalence values for persisting simulations under the strategies and time-points shown in Table 3 are reported in Table L in S1 Text.

All enhanced public health interventions are predicted to increase the potential for control of the imported strain. Increasing testing to the recommended level (RT) leads to a

**Table 3. Proportion of simulations in which the imported strain was detected and persisting, or extinct at 6 months, 2 years and 5 years post-importation.**

| Response strategies | Detected and Persisting (%) | | | Extinct (%) | | |
|---|---|---|---|---|---|---|
| | 6 months | 2 years | 5 years | 6 months | 2 year | 5 year |
| CT | 25.8 | 15.8 | 14.0 | 66.3 | 84.2 | 86.0 |
| CT+ PTT$_{R80}$ | 18.1 | 7.10 | 4.46 | 74.1 | 92.9 | 95.5 |
| CT+ PTT$_R$ | 15.5 | 4.3 | 1.44 | 76.7 | 95.7 | 98.6 |
| RT | 25.5 | 13.9 | 9.8 | 66.6 | 86.1 | 90.2 |
| RT +PTT$_{R80}$ | 17.8 | 3.98 | 0.02 | 74.3 | 96.0 | 100 |
| RT +PTT$_R$ | 15.5 | 1.34 | 0 | 76.6 | 98.7 | 100 |
| CT+ PTT$_{RC20}$ | 14.4 | 2.94 | 0.24 | 77.7 | 97.1 | 99.8 |
| CT+ PTT$_{RC30}$ | 13.7 | 2.28 | 0.1 | 78.4 | 97.7 | 99.9 |
| CT+ PTT$_{RC40}$ | 12.7 | 1.9 | 0.12 | 79.4 | 98.1 | 99.9 |
| CT+ PTT$_{RC50}$ | 12.3 | 1.36 | 0 | 79.8 | 98.6 | 100 |
| RT+ PTT$_{RC20}$ | 14.3 | 0.78 | 0 | 77.8 | 99.2 | 100 |
| RT+ PTT$_{RC30}$ | 13.4 | 0.44 | 0 | 78.7 | 99.6 | 100 |
| RT+ PTT$_{RC40}$ | 12.7 | 0.3 | 0 | 79.4 | 99.7 | 100 |
| RT+ PTT$_{RC50}$ | 12.2 | 0.18 | 0 | 79.9 | 99.8 | 100 |

**N**ote that the proportion of simulations where the imported strain was undetected and persisting was 7.9% at 6 months post-importation under all strategies and 0% at other time points.

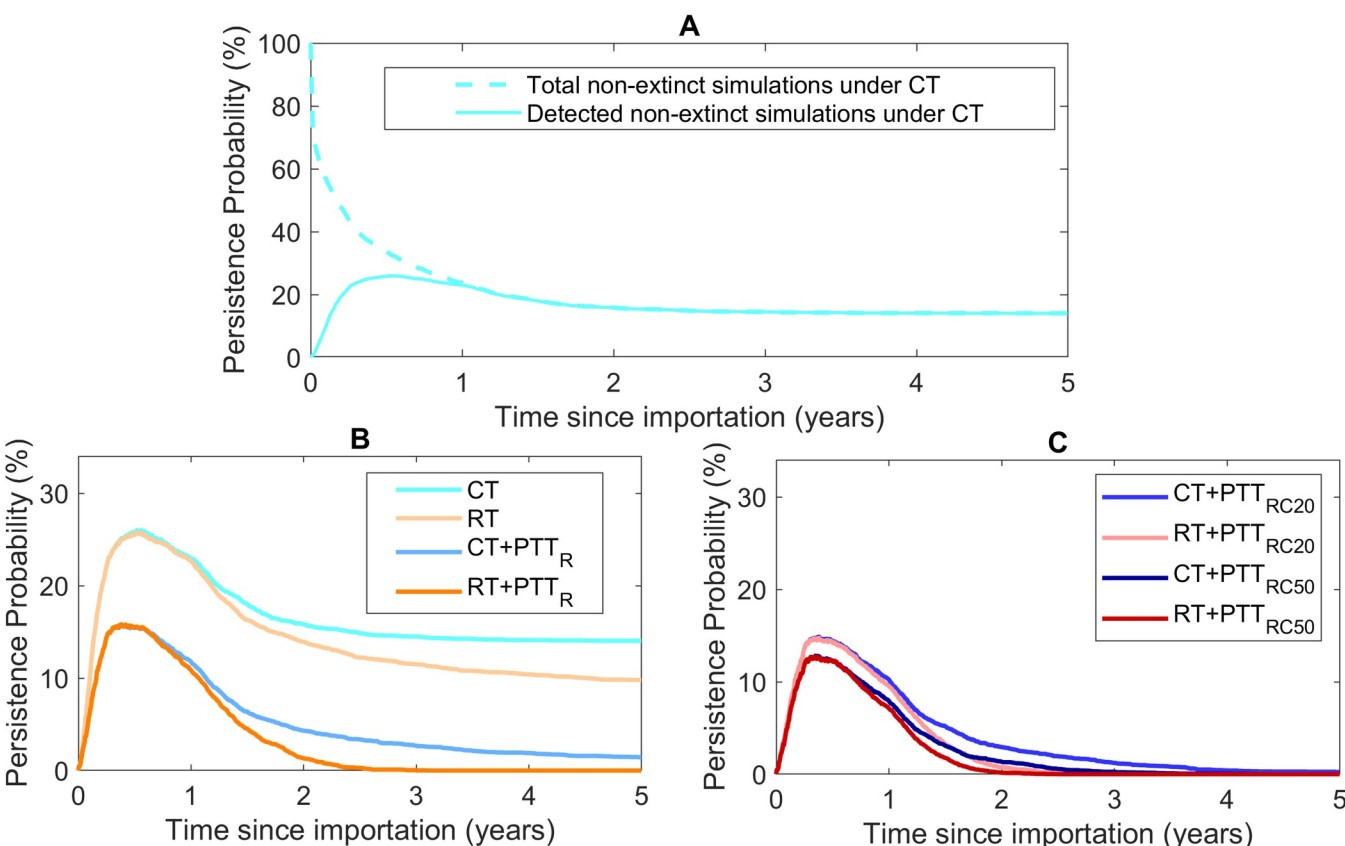

**Fig 3.** Panel A shows the persistence probability as a function of time under current testing. Dashed line: all simulations in which the imported strain persists, i.e., detected and undetected (100% at time = 0). Solid line: simulations in which the imported strain is detected and persisting (0% at time = 0). Dashed and solid lines converge at the end of the first year post-importation. Panel B shows trajectories for current and recommended testing with and without testing of regular partners. Panel C shows trajectories for current and recommended testing with testing of regular partners and a proportion of casual partners.

progressively greater impact than CT over time, with elimination of the imported strain rising from 84% to 86% and from 86% to 90% of simulations, at 2 years and 5 years, respectively. Further, this strategy greatly reduces the size of the subset of simulated outbreaks which persist, with mean prevalence of 0.57% (IQR [0.1%-0.84%]) and 1.9% (IQR [0.8%-2.9%]) at 2 and 5 years compared with 2.3% (IQR [0.5%-3.6%]) and 14.7% (IQR [14.3%-16.7%]) for CT at the same time points. When current testing is supplemented by testing and treating 100% of regular partners (CT+PTT$_R$), the probability of eliminating the imported strain increases to 96% and 98.5% of simulations and the mean prevalence of persisting strains is reduced to 0.25% (IQR [0.04%-0.38%]) and 0.45% (IQR [0.07%-0.78%]) at 2 and 5 years, respectively. The combination of recommended testing with testing/treating 100% of regular partners (RT+PTT$_R$) leads to elimination of the imported strain in 98.7% of simulations at 2 years, rising to 99.8% when supplemented by testing and treating 50% of casual partners in the last 2 months (RT+PTT$_{RC50}$). In both these strategies, the imported strain is eliminated in all 5000 simulations at 5 years post-importation.

Fig 3 provides a more detailed picture of detection and persistence over time for the interventions listed in Table 3. Panel A shows that the imported strain is either eliminated or detected within 12 months of importation in all simulations. Under all eight control strategies (panels B and C), the proportion of simulations in which outbreaks are detected but persist peaks at around 6 months post-importation. Inclusion of testing and treatment of regular

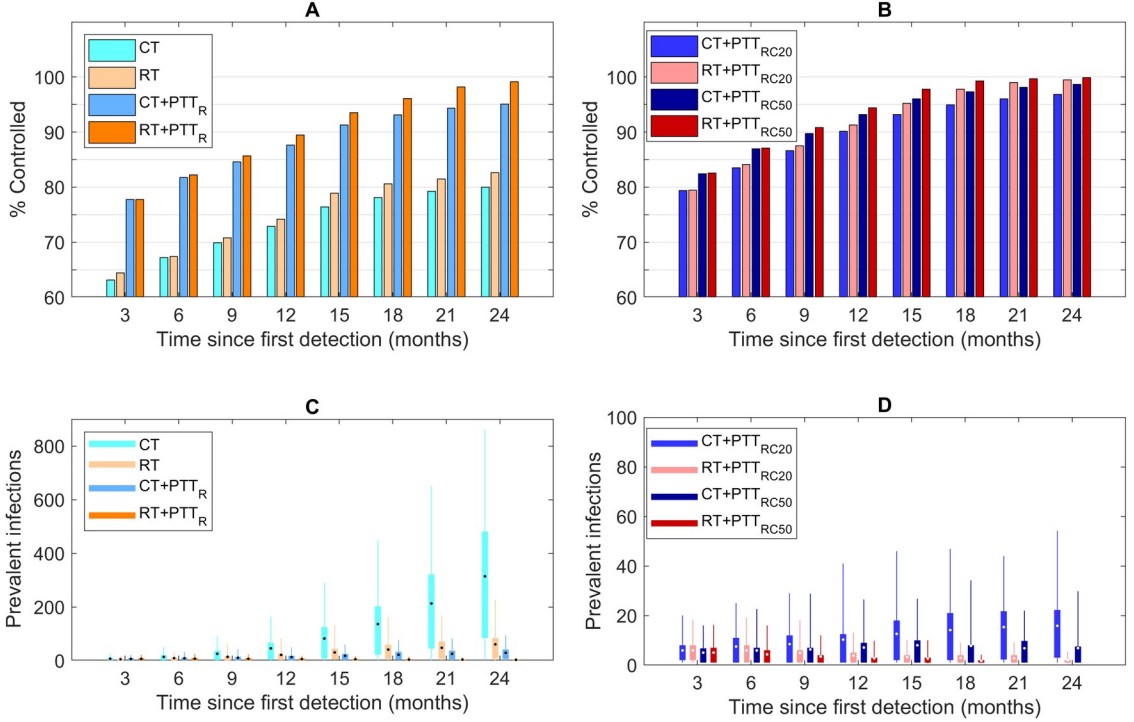

**Fig 4.** Panels A and B show the proportion of those simulations in which the imported strain is extinct, as a function of the time from the first detection and outbreak response strategy. Panels C and D show the outbreak size of imported NG strain for these simulations as a function of time from the first detection and outbreak response strategy. The box denotes the interquartile range (25% to 75%), the whiskers the quantile (5% to 95%), the horizontal line in each box the median, and the dot in each box the mean. Outbreak size is the number or infected individuals at each time point for simulations in which the imported strain persists.

partners greatly reduces the proportion of simulations in which the imported strain persists at each time point and this effect if further enhanced by testing/treatment of casual partners. The RT strategies have a modest impact alone but bring forward elimination of outbreaks when combined with partner treatment.

**Outbreak duration and containment.** In Fig 4 we show how success in eliminating the imported strain rises in quarterly periods after detection (panels A and B). Testing/treatment of regular partners is the most effective single strategy, increasing the probability of elimination within 3 months from 63% to 78% under CT when combined with partner treatment. This rises to 82% with testing/treatment of 50% of casual partners. To achieve >90% elimination at 12 months post-detection, RT in combination with testing/treatment of regular partners is required at a minimum. The probability of achieving elimination within 24 months of detection approaches 100% under strategies that additionally incorporate casual partner testing/treatment.

Panels C and D show how successful the various control strategies are in reducing the number of prevalent infections (outbreak size) at quarterly intervals. RT and/or addition of partner testing/treatment greatly reduce the growth in outbreak size in comparison to CT. Addition of RT to partner testing is more effective in constraining outbreak size than expanding testing and treatment to 20% or 50% of casual partners.

**Sensitivity analysis.** Sensitivity analysis results are provided in detail in section 3.2 of the S1 Text. These cover the impact of less favourable assumptions regarding diagnosis and treatment, the impact of increased condom-use and the effect of a 6-month delay in public health

response. Less favourable assumptions for diagnosis and treatment lead to the impact of the interventions being reduced (Table K in S1 Text). The outcomes from simulations were insensitive to changes in condom-use in the base-case scenario due to the dominant role of oro-pharyngeal transmission in our model. We also examined a scenario with a reduced symptomatic urethral proportion of 60% that, after recalibration, leads to a greater role of urethral transmission in sustaining prevalence. In this scenario increases in condom-use become an effective additional means of controlling NG outbreaks under CT (Table M in S1 Text). Finally, all interventions were much less effective if implemented at a 6-month delay post-importation (Table J in S1 Text).

## Discussion

In this modelling study we show that dissemination of imported strains of NG can be contained in a simulated Australian urban MSM population using moderately intense combinations of population-based and case-based testing/treatment strategies. The work is motivated by recent reports from Australia and the UK of imported XDR NG strains [5,7,16], including those with documented ceftriaxone/azithromycin dual treatment failure for oropharyngeal infection [7,8] and the potential for outbreaks that are difficult and expensive to control [19]. For instance, analysis of the historical rise in Australia of ciprofloxacin resistance [13] suggests that once resistant NG had become established overseas, variants of these resistant strains were repeatedly imported into Australia, leading eventually to a requirement to change the recommended treatment.

Under current testing practice, we find that around 1 in 7 imported NG strains will persist in the simulated MSM population at 5 years post-importation, with a mean prevalence of 15% at this point. However, both the probability of persistence and resulting prevalence can be greatly reduced by enhanced public health measures. When regular partners of NG cases are tested, as recommended in Australian contact tracing guidelines [17], the persistence probability drops to 0.5% at 5 years post-importation and falls further to 0.27% and 0%, respectively, if 20% or 50% of casual partners within the last 2 months can be tested/treated. These more intense case-based strategies are effective but need to be sustained for up to 2 years post-detection to eliminate >95% of imported strains. In the recent high-level azithromycin resistance NG outbreak in England [30], 118 total cases across both heterosexuals and MSM were investigated in an outbreak lasting more than 4 years. In this instance, ~35% of partners of heterosexuals but none of MSM were recorded as being tested, suggesting that reaching the simulated testing levels evaluated here may be challenging.

A complementary approach is to increase the rate of background STI testing. Current testing rates in Australia are high when compared with other international settings [31], but are substantially lower than recommended in the Australian STI Management [27] and STIGMA [28] guidelines. Our results show that this approach in isolation is more effective at constraining the size of outbreaks than eliminating them. In Australia, it may be plausible to achieve the recommended testing level given that in 2018 around 90% of surveyed Sydney MSM undertook an STI test at least once per year [23], and that a large proportion of high-risk MSM access three-monthly STI testing in conjunction with the uptake of HIV pre-exposure prophylaxis [32]. In addition, the new Australian STIGMA guidelines recommend quarterly testing for men with ~4+ partners per year [28], which would greatly increase the proportion of the MSM population undertaking three-monthly STI testing.

Combining case-based strategies with increased population testing is beneficial for control, with all such simulations in which greater than 20% of casual partners were tested/treated, resulting in elimination of the imported strain within 5 years. If 50% of casual partners are

tested/treated (RT+PTT$_{RC50}$), elimination of the imported strain occurs within 2 years of detection in all simulations with no outbreak exceeding 20 prevalent infections during this period. These results highlight the importance of combined approaches using both targeted (partner testing and treatment) and population-level strategies (increased community testing) to facilitate elimination. While not investigated here, synergistic effects between these interventions seem possible, with community testing reducing outbreak size and facilitating more intensive contact tracing.

In sensitivity analyses we also considered the effect of increasing condom use. This did not have a significant impact on control of the imported strain in the base-case. This is because under the base-case assumption that 90% of urethral cases are symptomatic and rapidly treated, urethral infection is relatively unimportant in overall transmission. However, if we assume a lower but potentially unrealistic symptomatic urethral infection proportion of 60% we found that increasing condom-use has additional benefit in controlling NG outbreaks. In the base-case, we assume that public health strategies are implemented immediately upon detection of an imported strain. When implementation was delayed by 6 months, partner therapy and increased community testing were much less effective.

The findings of this study should be interpreted in the context of the following limitations. The baseline (pre-intervention) model was calibrated to anatomical site-specific gonorrhoea prevalence as estimated by Zhang *et al*. [18], leading to inferred transmission probabilities, which qualitatively appear plausible but for which no definitive quantitative estimates are available. These prevalence targets were chosen over incidence estimates from the ACCESS study [25] and were similar to more data-driven estimates of prevalence from Victoria [33] but we note that the model produces similar site-specific incidence to 2018 ACCESS data. However, it is possible that rates of transmission may have risen since this time, as pre-exposure prophylaxis for HIV (PrEP) has been funded by the Pharmaceutical Benefits Scheme (PBS) since December 2017 with high uptake among eligible MSM, and evidence of potential changes in sexual risk behaviour associated with this rise according to GCPS Sydney 2018 [23]. The impact of PrEP on the spread of NG is unclear, as increases in condom-less intercourse may be balanced by requirements for regular STI testing for receipt of PrEP but future outbreaks may be more difficult to control than our simulations suggest. This study also considers just a single importation event, when in practice repeated importation events may occur. Though the model is able to incorporate multiple and concurrent importation events, further development is needed in regard to capture multiple concurrent strains and the capacity of public health responses to address simultaneous outbreaks. In addition, data to inform the rate at which imported NG infections occur is not currently available.

In regard to diagnostic sensitivity and treatment success, we compared our base-case results with more pessimistic assumptions, which resulted in lower extinction probabilities (except for RT and RT$_{STIGMA}$ at 5 years post initial importation), but typically slower outbreak growth. This occurs because as the same prevalence calibration targets were used for these simulations, assuming less effective treatment requires compensatory reductions in transmission probabilities compared to the base-case. Co-circulation of an endemic strain with the imported strain and any resultant interactions were not considered in this work, which focused on a single importation event. We note that there is limited evidence of mixed infections occurring at a single anatomical site but if this occurs it appears to be a rare occurrence [34]. Individual sexual practices are assumed not to change with age or time but given the simulation timeframe is just 5 years, in comparison to a sexual lifetime of ~50 years, we expect the impact of this on outcomes to be negligible. Age differences are implicitly captured to some extent by differences in partner-change rates, but may also occur in testing rates and the risk of overseas acquisition which are not considered here. Finally, we ignore bridging between MSM and heterosexual

populations which could increase the likelihood of an imported strain entering the MSM population.

In terms of technical advances over previous studies, our model reflects recent evidence of the importance of kissing as a transmission route (pharynx to pharynx) of NG in capturing the high observed prevalence and incidence of oropharyngeal infections. Secondly, we include group sex events as a subset of casual partnerships, reflecting real-world behavioural data. Finally, by explicitly linking to importations and integrating surveillance components, the model provides a framework for a more complete assessment of importation risk and strategies to control resistant NG infections.

From the perspective of public health, this is the first detailed exploration of the potential impacts of outbreak control strategies on the propagation of an imported NG strain. It suggests that a combination of increased case-based STI testing as well as more regular background testing would be successful in eliminating onward spread for a very high proportion of importations. Australian guidelines for management of gonococcal infections include a heightened response to cases of ceftriaxone and/or high-level azithromycin resistance [35]. This study provides additional evidence of the effect of these approaches alone and in combination with increases in background testing of asymptomatic MSM. Increased background testing has occurred in the context of the recent roll-out of pre-exposure prophylaxis for HIV in Australia [36] but our results do not support this being an effective strategy in isolation. We note that both feasibility and resource utilisation need further investigation, in particular the capacity to deal with multiple and potentially concurrent importation events, such as occurred in the emergence of ciprofloxacin resistance in Australia during the 1990s.

## Methods

An overview of the modelling methodology is provided here, with complete details provided in the S1 Text.

### Gonorrhoea natural history and transmission

The natural history of gonorrhoea is captured in a Susceptible-Exposed-Infectious-Recovered (SEIRS) framework. Individuals enter the population in the susceptible 'state', and can move progressively through the exposed (infected but not yet infectious) and infectious states following sexual contact with an infected person, before entering the recovered state (immune to reinfection) following resolution of infection, and then finally returning to the susceptible state. We assume a brief duration of immunity reflecting evidence of weak immunity following infection [37] but frequent reinfection [38]. Parameter values relating to gonorrhoea natural history have been derived, where possible, from published literature and are listed in Table 1.

Transmission is assumed to be possible between each pairing of anatomical sites leading to eight possible routes of transmission (See Table E in S1 Text). Infections at different anatomical sites within a given individual are assumed to be localised and independent, such that infection at one anatomical site does not influence the properties of infection (e.g., duration of infection) at any other anatomical site.

### Population and sexual behaviour

The model simulates a dynamic network of individuals connected via sexual partnerships, where both long- and short-term partnerships are considered, designated as 'regular' and 'casual', respectively. Individuals enter the sexually active population at age 16 years and on reaching age 65 years are replaced with a new individual aged 16 years, having the same sexual behaviour profile as the replaced individual.

On entry to the population, individuals are assigned a partner-type preference, which does not change over the simulation period. Preference can be for regular partners only, casual partners only, or both types of partner. Where relevant, individuals are assigned a casual partner acquisition rate (CPAR). Individuals are also assigned a positional preference for anal sex (receptive/insertive/no preference). Partnership durations for each partnership type are assigned at formation by sampling from a distribution (see Table B in S1 Text). Partner preferences and the CPAR distribution were based on data reported in the Gay Community Periodic Survey (GCPS) Sydney 2018 [23] and the Health In Men (HIM) Study [24], respectively. Group sex [39] is included as a subclass of casual partnership that only lasts for one day. Except for group sex, individuals can have at most one regular partner and/or one casual partner concurrently. An individual's sexual behaviour remains constant over time.

Within partnerships, individuals can engage in a variety of sexual acts (e.g., anal sex, oral-genital and oral-anal sex, kissing) that facilitate transmission. The frequencies of these acts during partnerships are based on data reported by Phang *et al*. [40] and Rosenberger *et al*. [41]. Kissing is an important transmission routine for oropharyngeal infection [42,43], and if it is not included, the model is not able to reproduce the high prevalence of oropharyngeal infection.

In the baseline model, individuals are tested for gonorrhoea, at a rate based on data reported in GCPS Sydney 2018 [23] and by the Australian Collaboration for Coordinated Enhanced Sentinel Surveillance of Sexually Transmissible Infections and Blood-borne Viruses (ACCESS) [26]. Condom use by partnership type is also derived from data reported in GCPS Sydney 2018 [23].

## Model calibration

The model was calibrated to estimated anatomical site-specific NG prevalence in a hypothetical community sample as reported in Zhang et al. [18]: oropharynx 8.6%; anorectum 8.3%; urethra 0.26%. This involved generating 10,000 parameter sets sampled from pre-specified ranges (Table I in S1 Text) and then determining the set with the smallest mean-squared difference between model-generated prevalence and the target values. This calibration establishes the baseline model representing the current situation, where gonorrhoea is endemic in the population, prior to the importation/emergence of a new strain. While the motivation for this work is the threat of new strains that are resistant to current first-line treatments, we are interested here in assessing the effectiveness of outbreak response strategies, assuming imported or emergent strains are still treatable with second-line drugs such as carbapenems [7].

## Simulation process

Results for each intervention strategy consist of 5,000 model simulations. Initially, each simulation is run for 10 years to establish the partnership network. After introduction of the imported strain, simulations are run with current testing rates until first detection of the imported strain, at which point the desired intervention is initiated and the simulation run for a further 5 years. Note that the imported strain can be detected when symptomatic patients seek treatment and asymptomatic patients are tested through screening.

At any time-point, a simulation is categorised as being in one of four states regarding the status of the imported strain: 1) *undetected and persisting*; 2) *undetected and extinct*; 3) *detected and persisting*; and 4) *detected and extinct*. All simulations are initiated in the *undetected* and *persisting* state, with transition to *detected and persisting* occurring at the first positive test and transition to *extinct* (*detected* or *undetected*) when the last case resolves.

## Supporting information

**S1 Text. Technical Appendix.** Table A. Parameters of gonorrhoea infection. Table B. Parameters of MSM sexual partnerships. Table C. Distribution of anal sex preference. Table D. Condom usage. Table E. Probability and frequency of sexual acts. Table F. Annual testing frequency for current and recommended testing levels by sexual activity category. Table G. Outbreak response strategies. Table H. Calibration targets of site-specific prevalence. Table I. Calibrated per-act transmission probabilities. Table J. Persistence probabilities at the 6 months and 2 and 5 years post importation (base). Table K. Persistence probabilities at the 6 months and 2 and 5 years post importation (pessimistic). Table L. Comparison of prevalence % at the 2 and 5 years post importation (base and pessimistic). Table M. Persistence probabilities at the 6 months and 2 and 5 years post importation (U). Fig A. Distribution of casual partner acquisition rate per 6 months (CPAR). Fig B. Distribution of population by different sexual partner preference in each year. The flat dashed lines are the data from GCPS Sydney 2018. Fig C. Comparison of number of casual partners acquired per 6 months: model output.vs HIM study. Fig D Site-specific prevalence among overall MSM. Fig E. The persistence probability as a function of time from importation of NG strain and response strategies. Fig F. Panel A and B show the proportion of those simulations in which the invading strain becomes extinct, as a function of the time from the first detection and outbreak response strategy. Panel C and D shows the outbreak size of invading NG strain among non-extinct simulations as a function of time from the first detection and outbreak response strategy. The box denotes the interquartile range (25% to 75%), the whiskers the quantile (5% to 95%), the horizontal line in each box the median, and the dot in each box the mean. Outbreak size is the number or infected individuals at each time point for simulations in which the invading strain persists. Fig G. The persistence probability as a function of time from importation of NG strain and response strategies. Fig H. Panel A and B show the proportion of those simulations in which the invading strain becomes extinct, as a function of the time from the first detection and outbreak response strategy. Panel C and D shows the outbreak size of invading NG strain among non-extinct simulations as a function of time from the first detection and outbreak response strategy. The box denotes the interquartile range (25% to 75%), the whiskers the quantile (5% to 95%), the horizontal line in each box the median, and the dot in each box the mean. Outbreak size is the number or infected individuals at each time point for simulations in which the invading strain persists. (DOCX)

## Author Contributions

**Conceptualization:** Chris Carmody, Basil Donovan, Rebecca J. Guy, John M. Kaldor, Monica M. Lahra, David A. Lewis, Michael Maley, Anna McNulty, Christine Selvey, David J. Templeton, David M. Whiley, David G. Regan, James G. Wood.

**Formal analysis:** Qibin Duan.

**Funding acquisition:** Chris Carmody, Basil Donovan, Rebecca J. Guy, Ben B. Hui, John M. Kaldor, Monica M. Lahra, Matthew G. Law, David A. Lewis, Michael Maley, Skye McGregor, David M. Whiley, David G. Regan, James G. Wood.

**Investigation:** David G. Regan, James G. Wood.

**Methodology:** Qibin Duan, Ben B. Hui, Matthew G. Law, David G. Regan, James G. Wood.

**Project administration:** David G. Regan, James G. Wood.

**Resources:** David G. Regan, James G. Wood.

**Software:** Qibin Duan.

**Supervision:** David G. Regan, James G. Wood.

**Validation:** Qibin Duan.

**Visualization:** Qibin Duan.

**Writing – original draft:** Qibin Duan.

**Writing – review & editing:** Chris Carmody, Basil Donovan, Rebecca J. Guy, Ben B. Hui, John M. Kaldor, Monica M. Lahra, Matthew G. Law, David A. Lewis, Michael Maley, Skye McGregor, Anna McNulty, Christine Selvey, David J. Templeton, David M. Whiley, David G. Regan, James G. Wood.

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
