## [Decision Letter · Decision Letter 0]

3 May 2021

Dear Dr. Wood,

Thank you very much for submitting your manuscript "Modelling outbreak response strategies for preventing spread of emergent Neisseria gonorrhoeae strains in men who have sex with men in Australia" for consideration at PLOS Computational Biology.

As with all papers reviewed by the journal, your manuscript was reviewed by members of the editorial board and by several independent reviewers. In light of the reviews (below this email), we would like to invite the resubmission of a significantly-revised version that takes into account the reviewers' comments.

We cannot make any decision about publication until we have seen the revised manuscript and your response to the reviewers' comments. Your revised manuscript is also likely to be sent to reviewers for further evaluation.

Sincerely,

Roger Dimitri Kouyos

Associate Editor

PLOS Computational Biology

Virginia Pitzer

Deputy Editor-in-Chief

PLOS Computational Biology

Reviewer's Responses to Questions

**Comments to the Authors:**

Reviewer #1: Review uploaded as an attachment

Reviewer #2: 1. The Introduction section must be more detailed. It must contain review of previous works

2.The authors must clearly state whether the model is new or whether they're modifying an existing model

3. The model equations should be presented

4. The results and discussion must clearly show how this model improved on previous ones.

Reviewer #3: In their manuscript, Duan et al use an individual-based, anatomical site-specific mathematical model of Ng transmission to ask the question: what types of testing and treatment strategies would be needed if an XDR Ng strain were introduced in the Australian MSM population? They highlighted several strategies that would be able to mitigate spread/eradicate circulation of this strain, namely combining case-based and population-based strategies (nicely summarized in Table 3). The model is well constructed and will certainly be useful for others examining specific questions on Ng transmission. There are, however, some concerns that need to be addressed.

The most obvious concern is how this model relates to the spread of XDR Ng. The authors assume that if such a strain is introduced, it can be treated just as well as any other Ng strain (ln 148-149). The real supposed concern with XDR Ng is that no treatment options will be available for these infections, meaning increased risk of spread. I do agree with the authors that other treatment alternatives could be used to cure Ng, but there will be delay in curing Ng in these individuals. The authors need to incorporate some aspects of treatment efficacy and duration of unsuccessful treatment if they want to infer anything on resistant variants circulating in the population. Furthermore, XDR Ng could naturally develop within the MSM population in Australia. Although rare, it could happen and might also affect the scenarios modeled. If the authors cannot address specific questions of resistance, they need to place their focus on Ng transmission rather than Ng resistance.

Another major assumption, from what I understand, is that only one introduction is expected and the resulting transmission patterns stem from this single introduction (ln 144-146). Given the worldwide network of MSM contacts and STI transmission, there would likely be several introductions of the XDR Ng strain in the population, which could have an impact on eradication of XDR Ng. These aspects also need to be considered in the model.

It is also unclear whether the model incorporates the fact that the prevalence of Ng infections in MSM has been steadily increasing. Similarly, condomless anal sex has also increased in several studies among MSM. The inputs used in this model are calibrated to 2018 levels, but does an epidemic background of increasing Ng prevalence and more frequent condomless anal sex render these strategies less effective?

Minor comments:

- ln 46. Should this be “current _testing and treatment_ practices”?

- ln 115. Are the authors really testing “surveillance” strategies? Is Ng required to be reported to the Australian government?

- ln 116-7. Not sure how the sentence “note that this…” relates to this paragraph.

- lns 168-170. How close are these inputs to the current situation?

- ln 177. The 2.3% prevalence refers to any site? Can this be divvied out by site?

- ln 361. By how much is the estimate missed when not included in this study?

- Figure 1. It would be helpful to have blocks regrouping events that are computed within the same algorithm.

- Table 1. Missing “(“ in the second column, row “Anorectum”.

- Technical appendix: Algorithm 1, step 3. Why does the % infected at the sites not match those listed on lines 132-133?

- Technical appendix: There are two “Algorithm 3” in the manuscript.

- I would highly recommend the authors place the data and programming code on a publicly available server.

**Have all data underlying the figures and results presented in the manuscript been provided?**

Reviewer #1: Yes

PLOS authors have the option to publish the peer review history of their article (what does this mean?). If published, this will include your full peer review and any attached files.

Reviewer #1: No

Reviewer #2: No

Reviewer #3: No

**Have the authors made all data and (if applicable) computational code underlying the findings in their manuscript fully available?**

Reviewer #2: **No: **

Reviewer #3: **No: **I cannot find the link to any code.
---

## [Decision Letter · Decision Letter 1]

26 Aug 2021

Dear Dr. Wood,

We are pleased to inform you that your manuscript 'Modelling response strategies for controlling gonorrhoea outbreaks in men who have sex with men in Australia' has been provisionally accepted for publication in PLOS Computational Biology.

Best regards,

Roger Dimitri Kouyos

Associate Editor

PLOS Computational Biology

Virginia Pitzer

Deputy Editor-in-Chief

PLOS Computational Biology

Reviewer's Responses to Questions

**Comments to the Authors:**

Reviewer #2: The paper can be accepted

Reviewer #3: I thank the authors for their clear responses and careful consideration of my comments.

One remaining issue is that XDR Ng strains are still the major focus of the introduction (and in one small part of the discussion). The focus needs to be placed on the imported infection, with XDR being an important example. It would be nice to add, to the discussion, how the model could be fine-tuned to address other research questions on the imported strain (e.g. XDR, one that might be more transmissible, etc.). Addressing this comment would only require adding some sentences and a slight restructuring of paragraphs.

Reviewer #4: No further comments

**Have the authors made all data and (if applicable) computational code underlying the findings in their manuscript fully available?**

Reviewer #2: Yes

Reviewer #3: Yes

Reviewer #4: Yes

PLOS authors have the option to publish the peer review history of their article (what does this mean?). If published, this will include your full peer review and any attached files.

Reviewer #2: No

Reviewer #3: No

Reviewer #4: No

---

## [Editor Report · Acceptance letter]

29 Oct 2021

PCOMPBIOL-D-21-00242R1 

Modelling response strategies for controlling gonorrhoea outbreaks in men who have sex with men in Australia

Dear Dr Wood,

I am pleased to inform you that your manuscript has been formally accepted for publication in PLOS Computational Biology. Your manuscript is now with our production department and you will be notified of the publication date in due course.

With kind regards,

Olena Szabo
